# Vision-Language Interpreter for Robot Task Planning

Keisuke Shirai[1*], Cristian C. Beltran-Hernandez[2], Masashi Hamaya[2], Atsushi Hashimoto[2],
Shohei Tanaka[2], Kento Kawaharazuka[3], Kazutoshi Tanaka[2], Yoshitaka Ushiku[2], Shinsuke Mori[4]

*Abstract*— Large language models (LLMs) are accelerating the development of language-guided robot planners. Meanwhile, symbolic planners offer the advantage of interpretability. This paper proposes a new task that bridges these two trends, namely, *multimodal planning problem specification*. The aim is to generate a problem description (PD), a machine-readable file used by the planners to find a plan. By generating PDs from language instruction and scene observation, we can drive symbolic planners in a language-guided framework. We propose a Vision-Language Interpreter (ViLaIn), a new framework that generates PDs using state-of-the-art LLM and vision-language models. ViLaIn can refine generated PDs via error message feedback from the symbolic planner. Our aim is to answer the question: How accurately can ViLaIn and the symbolic planner generate valid robot plans? To evaluate ViLaIn, we introduce a novel dataset called the problem description generation (ProDG) dataset. The framework is evaluated with four new evaluation metrics. Experimental results show that ViLaIn can generate syntactically correct problems with more than 99% accuracy and valid plans with more than 58% accuracy. Our code and dataset are available at `https://github.com/omron-sinicx/ViLaIn`.

## I. INTRODUCTION

Natural language is a prospective interface for non-experts to instruct robots intuitively [1]–[3]. Earlier studies have used recurrent neural networks [4], [5] to map abstract linguistic instructions to representations for robots [1], [6], [7]. Here, the linguistic instructions represent desired goal conditions. More recent studies use large language models (LLMs) [8]–[10] to directly generate robot plans from the instructions [11]–[14]. These language-guided planners utilize few-shot prompting to solve tasks without training [15]. The plans are a sequence of discrete symbolic actions (e.g., `pick(a)` and `place(a, b)`) that complete the task. We aim to strengthen the language-guided planners in terms of the improvement of interpretability.[1] Interpretability is essential to gain the trust of the user and provide insights into the robot's decision-making process [16]. For example, the

[1]Keisuke Shirai and Shinsuke Mori are with Kyoto University, Kyoto 606-8501, Japan (email: [shirai.keisuke.64x@st.kyoto-u.ac.jp,forest@i.kyoto-u.ac.jp])

[2]Cristian C. Beltran-Hernandez, Masashi Hamaya, Atsushi Hashimoto, Shohei Tanaka, Kazutoshi Tanaka, Yoshitaka Ushiku are with the OMRON SINIC X Corporation, Tokyo 113-0033, Japan (email: [cristian.beltran, masashi.hamaya, atsushi.hashimoto, shohei.tanaka, kazutoshi.tanaka, yoshi-taka.ushiku]@sinicx.com)

[3]Kento Kawaharazuka is with the University of Tokyo, 73-1 Hongo, Bunkyo-ku, Tokyo, 113-8656, Japan (email: kawaharazuka@jsk.t.u-tokyo.ac.jp)

*Work was done while the first author was an intern at OMRON SINIC X Corporation.

[1]We define interpretability as a mechanism to provide insights into the inner workings of the system.

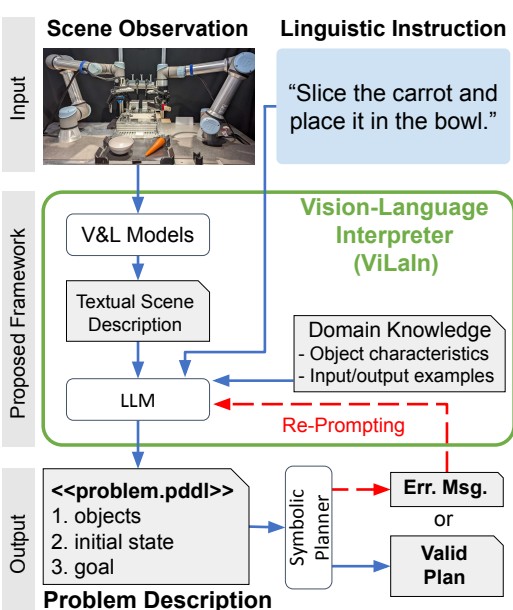

**Scene Observation**     **Linguistic Instruction**

Fig. 1. Overview of our approach. The vision-language interpreter (ViLaIn) generates a problem description from a linguistic instruction and scene observation. The symbolic planner finds an optimal plan from the generated problem description.

identification of failure causes through interpretation leads to continuous improvement of overall performance.

Robot task planning has traditionally been solved using symbolic planning [17]. Modern symbolic planners use the Planning Domain Definition Language (PDDL) to describe planning problems. In PDDL, a planning problem is defined in two parts: the *domain* that defines the state of variables and actions, and a *problem description* (PD) that defines the objects of interest, their initial state, and the desired goal state [18], [19]. The domain and problem are inputs to the planner to find a plan, a sequence of symbolic actions.

Symbolic planners offer several benefits. The domain and problem descriptions are human-readable, especially when variable names are chosen intuitively. Moreover, the obtained plans are guaranteed to be logically correct. Considering these advantages, combining symbolic planning and language-guided planning is a promising research direction to realize interpretable robots. To that end, we proposed generating the PDs from natural language instructions. Since the linguistic instructions only represent the goal conditions, additional information about the environment is required to generate the initial state (e.g., an image representing the current environment). We refer to this additional information

| Approach | Input other than linguistic instruction | Output |
|---|---|---|
| Huang et al. [11] | — | Symbolic action |
| Raman et al. [12] | — | Symbolic action |
| Text2Motion [13] | PDDL scene desc. | Symbolic action |
| SayCan [23] | Image | Pre-defined skill |
| RT-2 [24] | Image | Low-level action |
| ProgPrompt [14] | — | Program code |
| Code as Policies [3] | Image | Program code |
| LLM+P [25] | Linguistic scene desc. | Problem desc. |
| ViLaIn (ours) | Image | Problem desc. |

as *scene observations*.

We tackle the *multimodal planning problem specification* task, a new task for transforming linguistic instructions and scene observations into logically and semantically correct PDs. The PDs have to be executable by the symbolic planners. This paper investigates how accurately we can generate such PDs with a state-of-the-art LLM [9] and vision-language model [20], [21] without additional training. We propose a Vision-Language Interpreter (ViLaIn), a new framework to solve the PD generation task, illustrated in Fig. 1. ViLaIn consists of three modules that generate each part of the PDs. The complete PD is assembled by concatenating these parts. Furthermore, ViLaIn can refine the generated PDs via error feedback from the symbolic planner. The planner uses a pair of the generated PD and the domain description to find a plan. We use Fast Downward [22] as the symbolic planner throughout this paper.

To evaluate ViLaIn, we introduce a novel dataset called the problem description generation (ProDG) dataset. The ProDG dataset consists of linguistic instructions, scene observations, and domain and problem descriptions. The descriptions are written in PDDL [19]. This dataset covers three domains: cooking as a practical robot domain, and the blocks world and the tower of Hanoi as classical planning domains. We propose four new evaluation metrics to evaluate ViLaIn from multiple perspectives.

## II. RELATED WORK

This section describes previous work on language-guided planning, symbolic planning, and scene recognition in computer vision. Table I summarizes the difference between several studies mentioned here and ViLaIn.

### A. Planning from Natural Language

Task planning from natural language has been actively studied [11], [20], [23]. Converting linguistic instructions into symbolic actions via neural networks is a typical approach [7], [26]. More recent studies [11]–[14] use LLMs and directly generate plans with few-shot prompting [15]. However, these language-guided planners have two issues. First, their systems hide the inner workings by generating plans end-to-end. Second, the obtained plans are not guaranteed to be logically correct. ViLaIn resolves these issues by converting instructions into human-readable PDs and driving symbolic planners to find plans with the generated PDs. A

recent study uses LLMs to convert linguistic instructions and images into programs to complete robot tasks [3]. PDs describe tasks more specifically, and their logical correctness is automatically verifiable. In other words, ViLaIn has the potential to deliver validated machine-readable information to other language-guided planners as an auxiliary input.

More recent studies have used LLMs to convert natural language inputs to PDs [25], [27]. However, one study [25] assumes that scene descriptions (the objects and initial state) are provided in natural language, which is not practical for real applications. Another work [27] focuses on only generating the goal specifications. Contrary to these studies, ViLaIn uses images for scene descriptions and generates the whole PDs, including the objects and initial states.

### B. Symbolic Planning with PDDL

Symbolic planning (automated planning) has been used to solve robotic tasks [17]. Symbolic planners [22], [28] use domain and problem descriptions to find plans, which are sequences of (symbolic) actions that alter the environment from its initial state to a goal state. The descriptions are written in formal languages, such as PDDL [19] and PDDLStream [29]. Robots execute low-level actions based on the found high-level plans of PDDL [30]–[32]. This framework enables robots to solve various problems but assumes a preparation of corresponding PD for each problem. ViLaIn is designed to collaborate with those PDDL-based planning frameworks by translating linguistic instructions into PDs.

### C. Scene Recognition for Planning Problem Specification

The object part of PDs lists objects required for the task. This work generates the objects from scene observations. This can be viewed as object detection in computer vision. Classical object detectors [33], [34] have been developed focusing on a fixed number of classes (e.g., person and dog). However, our task handles objects not included in the classes. Hence, we use an open-vocabulary object detector [20], [35]. These detectors have recently gained attention because they can detect arbitrary objects using text queries.

The *initial state* represents object relationships and their states. Detecting such scene descriptions from images has been addressed on visual relationship detection [36], [37] or scene graph generation [38], [39]. Previous work trained a model with PDDL predicates and demonstrated it in real robot domains [40]. We use a state-of-the-art LLM and vision-language model to generate the initial state.

## III. PROBLEM STATEMENT

We focus on multimodal planning problem specification, a new task for bridging language-guided planning and symbolic planning. The input is a quadruple $(L, S, D_D, D_K)$; a linguistic instruction $L$, a scene observation $S$, a domain description $D_D$, and domain knowledge $D_K$. $D_D$ defines parts common to all problems: object types, predicates, and symbolic actions. $D_K$ supports $D_D$ by providing more specific information on each problem, such as object characteristics (e.g., the cutting board is round, the counter is black) and actual input/output examples.

The output is a PD $P$ consisting of $(O, I, G)$. The objects $O$ consist of objects required for the task completion (e.g., *carrot* and *knife*). The initial state $I$ consists of a set of propositions that represent the initial state of the environment (e.g., `(at carrot counter)`). A proposition is formed by providing a predicate with arguments. For example, providing a predicate `(at ?a1 ?a2)` with `(a1, a2)` `= (carrot, cutting_board)` forms a proposition `(at carrot cutting_board)` meaning "the carrot is at the cutting board." The goal specification $G$ consists of a set of propositions that represent the desired goal condition of the environment. For example, `(and (at carrot bowl) (is-sliced carrot))` represents the goal condition that "the carrot should be sliced and should be at the bowl." $P$ and $D_D$ are written in PDDL [19], following previous work [25], [27]. We refer to $O$, $I$, or $G$ with PDDL (e.g., the PDDL objects). The goal of this task is obtaining a function $M : (L, S, D_D, D_K) \rightarrow (O, I, G)$. $P$ must be machine-readable and executable by the symbolic planner.

## IV. VISION-LANGUAGE INTERPRETER

ViLaIn consists of three modules: the object estimator, the initial state estimator, and the goal estimator. We describe these modules in this section.

### A. Object Estimator

The PDDL objects $O$ list objects of interest in the scene observations $S$. However, the observed objects vary greatly from domain to domain. Further, it must recognize various objects that classical object detectors cannot handle. For this reason, we use Grounding-DINO [20], a state-of-the-art open-vocabulary object detector. We assume that the list of objects for the task is known. The object list can be used as the text query. However, we found from preliminary experiments that simply using the object list fails to detect several objects. To address this issue, we elaborate the query using the domain knowledge (e.g., "cutting board" $\rightarrow$ "round cutting board" and "knife" $\rightarrow$ "kitchen knife").[2] In our setting, these elaborated queries are included in the domain knowledge $D_K$. The detected objects are converted into a PDDL format by rules.

### B. Initial State Estimator

The PDDL initial states $I$ must specify the initial state of the environment using propositions. Here, different predicates from $D_D$ should be used for different domains to represent the propositions. In addition, omitting a single proposition could cause an invalid PD by making reaching the goal from the initial state impossible. We implement the initial state estimator with a combination of an LLM and image captioning model. We use BLIP-2 [21] as the captioning model and GPT-4 [9] as the LLM. BLIP-2 generates captions for the objects specified by bounding boxes. GPT-4 uses few-shot prompting and leverages input/output examples in $D_D$ to derive available predicates.

[2]In this work, we assume that the domain knowledge is created by humans, and we leave the automatic generation of it to future work.

### C. Goal Estimator

The PDDL goal specifications $G$ must represent the desired goal conditions specified by the linguistic instructions $L$. Generating $G$ requires $O$ to refer to the object list and $I$ to consider the relationships of the objects. We implement the goal estimator with an LLM, following previous work [13], [27]. We adopt GPT-4 to generate $G$ from $L$, $O$ using the few-shot prompting.

### D. Corrective Re-Prompting

Planning based on generated PDs might fail if they are syntactically incorrect or contain erroneous propositions. In such cases, the planner stops planning and returns an error message, a clue to refine the erroneous parts. ViLaIn can recover from the error by re-prompting GPT-5 to refine the PD. We refer to this technique as Corrective Re-prompting (CR), following previous work [12]. The prompt consists of input/output examples in $D_K$, the current input ($L$ and $S$), the generated problem $P$, and the error message. We also use Chain-of-thought (CoT) prompting [41]–[43] to further strengthen CR. CoT is a technique to solve complex reasoning tasks using LLMs. In CR with CoT, ViLaIn first generates an explanation for the error message, adds it to the prompt, and then re-generates the PD based on the prompt. Note that ViLaIn performs CR with CoT only if the planner returns an error message.

## V. DATASET

To evaluate ViLaIn, we created a novel dataset called the ProDG dataset that consists of linguistic instructions, scene observations, and PDDL domains and problems. This dataset covers three domains: cooking, the blocks world (Blocksworld), and the tower of Hanoi (Hanoi). **Cooking** is a simplified task of making a salad. It only considers slicing vegetables and placing them in the bowl. $O$ and $I$ represent the location and state of vegetables and tools in $S$, and $G$ describes the post-action state of food(s) specified by $L$. **Blocksworld** is a classical planning domain [44], and it aims to stack blocks according to $L$. $O$ and $I$ represent the relative position of blocks in $S$, and $G$ describes the final position of the blocks. **Hanoi** is a classical planning domain [45], and it aims to stack disks while keeping constraints. $O$ and $I$ represent the relative position of disks in $S$, and $G$ describes the final position of the disks.

Each domain has one domain description and ten PDs. Each problem has one linguistic instruction and one scene observation. Fig. 2 shows examples of linguistic instructions $L$ and scene observations $S$. For the Hanoi domain, $L$ is identical through all problems. This aims to investigate whether ViLaIn can generate different $G$ based on $O$ and $I$. The descriptions for the cooking domain were created from scratch, while those for the Blocksworld and Hanoi domains were created based on the PDDL files in pddlgym [46]. We confirmed that all the created PDs are syntactically correct and have solutions using Fast Downward [22] and VAL, a plan validation software.[3]

[3]https://github.com/KCL-Planning/VAL

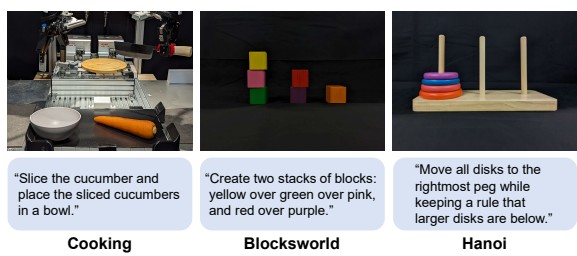

| "Slice the cucumber and place the sliced cucumbers in a bowl." | "Create two stacks of blocks: yellow over green over pink, and red over purple." | "Move all disks to the rightmost peg while keeping a rule that larger disks are below." |
| :---: | :---: | :---: |
| **Cooking** | **Blocksworld** | **Hanoi** |

Fig. 2. Examples of scene observations and linguistic instructions.

## A. Evaluation Metrics

In PD generation, previously proposed metrics roughly calculate the planning success rate or are domain-specific ones [25], [27]. It would be ideal to have metrics that evaluate PDs from multiple perspectives regardless of domain. To this end, we introduce new metrics: $R_{syntax}$, $R_{plan}$, $R_{part}$, and $R_{all}$.

***$R_{syntax}$:*** PDs must be syntactically correct. $R_{syntax}$ calculates the ratio of such PDs. A PD is considered to be syntactically correct if VAL returns no warnings and exit codes for a pair of the domain and the generated PD.

***$R_{plan}$:*** Even if the PDs are syntactically correct, they might not have valid plans due to incorrect objects in $O$ and incorrect or contradictory propositions in $I$ and $G$. $R_{plan}$ calculates the ratio of the PDs having valid plans. The plans are obtained using Fast Downward [22]. A plan is considered to be valid if VAL returns no error messages.

***$R_{part}$ and $R_{all}$:*** Different from the above two metrics, $R_{part}$ and $R_{all}$ evaluate how close the generated PDs are to the ground truth ones. $R_{part}$ calculates the recall of the problem parts between the files. $R_{part}$ is independently computed for $O$, $I$, and $G$ based on object labels and propositions. Unlike $R_{part}$, $R_{all}$ calculates the ratio of problems containing all the ground truth object labels and propositions.

## VI. EXPERIMENTS

We conduct experiments to investigate how accurately ViLaIn can generate PDs on the ProDG dataset. This section first describes the generation settings of ViLaIn and then discusses experimental results.

## A. Generation Settings of ViLaIn

GPT-4 used few-shot prompting with three input/output examples in the same domain as the current task. ViLaIn can refine erroneous PDs by CR $n$ times. PDs with corrected grammatical errors can still have semantic errors, causing no valid solutions. In such cases, CR should be performed at least twice. Thus, we set $n$ to two. For evaluation, we generated ten PDs per problem by varying the example combinations. The resulting 100 problems per domain are used to evaluate ViLaIn.

## B. Evaluation of Generation Results by ViLaIn

Table II shows the results. The $R_{syntax}$ scores are more than 99% in all the three domains. This means that ViLaIn can generate syntactically correct PDs for these domains utilizing the three input/output examples. The $R_{plan}$ scores indicate

TABLE II
PERFORMANCE ON THE PRODG DATASET

| Domain | $R_{syntax}$ | $R_{plan}$ | $R_{part}$ | | | $R_{all}$ |
| :--- | :---: | :---: | :---: | :---: | :---: | :---: |
| | | | $O$ | $I$ | $G$ | |
| Cooking | 0.99 | 0.99 | 1.00 | 0.93 | 0.93 | 0.71 |
| Blocksworld | 0.99 | 0.94 | 0.98 | 0.79 | 0.89 | 0.36 |
| Hanoi | 1.00 | 0.58 | 0.89 | 0.46 | 0.33 | 0.12 |

that 94% or more PDs have valid plans in the cooking and Blocksworld domains. However, in the Hanoi domain, the $R_{plan}$ score is only 58% due to its challenging setting. We found from the outputs that ViLaIn tends to omit some propositions in this domain, making the PDs invalid.

For $R_{part}$, the scores on $I$ and $G$ are smaller than those on $O$. This implies that generating $I$ and $G$ is more challenging than $O$. We found that mistakenly detected objects cause this. Predicates such as `on` or `at` take two objects as arguments. Propositions created with the predicates and mistakenly detected objects affect other propositions. For example, `(on red_block blue_block)` can be `(on red_block green_block)` `(on green_block blue_block)` with a mistakenly detected `green_block`, making them all incorrect propositions. We consider that generating these incorrect propositions causes such results.

Finally, the $R_{all}$ score is 71% in the cooking domain, 36% in the Blocksworld domain, and 12% in the Hanoi domain. The scores in the cooking and Hanoi domains make sense considering the $R_{plan}$ and $R_{part}$ scores. However, the score is unexpectedly low in the Blocksworld domain. We found that PDs in the Blocksworld domain tend to contain a few incorrect propositions of block relationships. In some cases, the block positioning is mistakenly reversed (e.g., `(on blue_block red_block)` `(on red_block green_block)` is reversed to `(on green_block red_block)` `(on red_block blue_block)`). We consider that these lead to the low $R_{all}$ score in this domain.

## VII. CONCLUSION

We have proposed Vision-language interpreter (ViLaIn) to tackle multimodal planning problem specification. To evaluate ViLaIn, we have introduced a novel dataset called the problem description generation (ProDG) dataset with new metrics. The experimental results have shown that ViLaIn can generate syntactically correct PDs and more than half of the PDs have valid plans. Interesting future directions include (i) constructing a robotic system with ViLaIn that executes linguistic instructions, (ii) refining PDs via errors from real robots, and (iii) reducing human effort for new tasks.

## ACKNOWLEDGMENT

We would like to thank Hirotaka Kameko for his helpful comments. This work was supported by JSPS KAKENHI Grant Number 20H04210 and 21H04910 and JST Moonshot R&D Grant Number JPMJMS2236.

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
