# OpenReview forum: "Vision-Language Interpreter for Robot Task Planning"
_IEEE.org/2024/ICRA/Workshop/CookingRobot — CookingRobot2024 Poster_

### Official Review · Reviewer_Jws6 · 2024-04-15
**Review for "Vision-Language Interpreter for Robot Task Planning"**

**Rating:** 8
**Confidence:** 4

**Review:**

The paper proposes ViLaIn, a framework for generating a problem description file from task instruction and visual observation utilizing the Vision-Language model and LLM. The output of ViLaIn is used for planning with PDDL. In the experiment, the plan is evaluated with a planning dataset (ProDG), which includes planning of the cooking task.

Comments on the Video
- The video fully illustrates the problem setting and the proposed method, which is very helpful for readers to understand the work.
- I would like to see the experiment results (and especially the failure cases) in the video, though it is reasonable that the video does not have real robot experiments in cooking since the work focuses on planning.

---

### Official Review · Reviewer_Y5Ch · 2024-04-16
**The review for "Vision-Language Interpreter for Robot Task Planning"**

**Rating:** 8
**Confidence:** 5

**Review:**

This paper proposes a framework for generating a problem description for the symbolic planner from linguistic instructions and scene observations using LLM and vision-language models. Action planning adapted to the actual environment from linguistic instructions is a very important issue in cooking where recipes exist, and the results of this paper should improve workshop discussions on this important issue.

Major comment:
* Regarding evaluation, in order to realize a robot system that executes linguistic instructions with ViLaIn, it is necessary to evaluate whether or not the task plan is actually executable and achieves the instructions when the plan is made with the inferred PD. Is such evaluation included in the proposed metrics?

Vieo:
* This video provides a better understanding of this research. It would be great if there was a video of the cooking process using the task planning with the proposed ViLaIn, or the result of the task planning which is the same as the procedure in the first slice video.